# Naturalistic development of trait mindfulness: A longitudinal examination of victimization and supportive relationships in early adolescence

Michael T. Warren [1]*, Kimberly A. Schonert-Reichl[1], Randip Gill[1], Anne M. Gadermann[1,2], Eva Oberle[1]

1 Human Early Learning Partnership, School of Population and Public Health, University of British Columbia, Vancouver, British Columbia, Canada, 2 Centre for Health Evaluation and Outcomes Sciences, Providence Health Care Research Institute, Vancouver, British Columbia, Canada

* michael.warren1936@gmail.com

## Abstract

Scholars have only just begun to examine elements of young adolescents' social ecologies that explain naturalistic variation in trait mindfulness and its development over time. We argue that trait mindfulness develops as a function of chronically encountered ecologies that are likely to foster or thwart the repeated enactment of mindful states over time. Using data from 4,593 fourth and seventh grade students (50% female; $M_{ageG4}$ = 9.02; 71% English first language) from 32 public school districts in British Columbia (BC), Canada, we examined links from peer belonging, connectedness with adults at home, and peer victimization to mindfulness over time. Variable-centered analyses indicated that young adolescents with lower victimization in fourth grade reported higher mindfulness in seventh grade, and that cross-sectionally within seventh grade victimization, peer belonging, and connectedness with adults at home were each associated with mindfulness. Contrary to our hypothesis, connectedness with adults at home moderated the longitudinal association between victimization and mindfulness such that the negative association was stronger among young adolescents with high (vs. low) levels of connectedness with adults at home. Person-centered analysis of the fourth graders' data confirmed our variable-centered findings, yielding four latent classes of social ecology whose mindfulness levels in seventh grade largely tracked with their victimization levels (from highest to lowest mindfulness): (1) *flourishing relationships*, (2) *unvictimized but weak relationships with adults*, (3) *moderately victimized but strong relationships*, and (4) *victimized but strong relationships*. Overall, our findings contribute to a growing body of evidence indicating that trait mindfulness may develop as a function of ecologically normative experiences in young adolescents' everyday lives.

**Data Availability Statement:** Data are available upon request from Population Data BC: https://

www.popdata.bc.ca/data. Data access is subject to approval by the Data Stewards representing the Human Early Learning Partnership, for ethical and privacy reasons, because the data pertain to individuals. In alignment with regulations for accessing population level data in Canada, data can only be accessed through Population Data BC's Secure Research Environment cloud server. We enjoyed no special access privileges in accessing data. Data set name: MDI_g4_2010to 2016_g7_2013to2018_12298_2018_8_21. Variable names: SD_gend (gender), firstlang_g4 (first language), q96_g7 - q98_g7 (mindfulness in 7th grade), q53_g4 - q56_g4 (victimization in 4th grade), q53_g7 - q56_g7 (victimization in 7th grade), q28_g4 - q30_g4 (connectedness to adults at home in 4th grade), q28_g7 - q30_g7 (connectedness to adults at home in 7th grade), q37_g4 - q39_g4 (peer belonging in 4th grade), q37_g7 - q39_g7 (peer belonging in 7th grade), q75_g4 - q77_g4 (self-regulation in 4th grade), q93_g7 - q95_g7 (self-regulation in 7th grade), Collection_Year_g4 (cohort).

**Funding:** The authors received no specific funding for this work.

**Competing interests:** The authors have declared that no competing interests exist.

## Introduction

A growing body of research documents the benefits of trait mindfulness—a person's characteristic acceptance and awareness of present-moment experience—both in positive functioning and as a protective factor in the context of risks to poor functioning during adolescence [1–6; but compare this to 7]. As such, trait mindfulness has emerged as a viable internal developmental asset, the developmental course of which has recently become a focus of scholarly attention [8–12]. However, scholars have only just begun to examine elements of adolescents' social ecologies that explain naturalistic variation in trait mindfulness and its development over time. We argue that, as an entrenched disposition, trait mindfulness develops as a function of recurring experiences that are likely to foster or thwart the repeated enactment of mindful states over time [13, 14]. In this study, we examine the roles of one risk factor (peer victimization) and two assets (peer belonging and connectedness with adults at home) in predicting subsequent levels of trait mindfulness 3 years later during early adolescence.

### Development of trait mindfulness

Mindfulness has been described in various ways, with definitions centering on acceptance and awareness of experience as it unfolds in the present moment. For example, Brown and colleagues have defined mindfulness as "receptive attention to and awareness of present events and experiences" p. 212 [15]. *Trait* mindfulness therefore entails one's characteristic level of mindfulness across a wide variety of situations and over time [16, 17], whereas *state* mindfulness refers to context dependent (and thus unstable) enactments of mindfulness [18, 19] over relatively brief periods of time. Although there is general consensus that traits exhibit firm rank-order stability over time, a robust body of evidence shows that traits can and do develop throughout the lifespan [20, 21] and there is reason to believe that during adolescence traits are more malleable than in later periods of development [21–23].

Based on modern dynamic personality trait theories [14, 24], we argue that trait mindfulness develops, in part, as a function of recurring experiences that give rise to the repeated instantiation of mindful states. For example, in an 8-week mindfulness intervention with an adult sample, growth in weekly state mindfulness experienced during meditation predicted increases in trait mindfulness from pre-test to post-test [13]. Similarly, in a study that gave 4th and 5th graders opportunities to practice mindful states three times per day for 12 weeks, the mindfulness group (relative to a business as usual control group) displayed improvement in trait mindfulness from pre-test to post-test [25].

This general principle may extend beyond mindfulness interventions and into the contexts in which adolescents are embedded in their everyday lives at school, at home, with friends, and so on. Based on the TESSERA model of personality development [14], which holds in part that regularly encountered situations repeatedly trigger states that collectively comprise a trait, ecologies that are likely to repeatedly enable mindful states may contribute to the development of trait mindfulness. Conversely, ecologies likely to repeatedly trigger mindless responses (e.g., rumination, anxiety, social comparison) should impede the development of trait mindfulness. Initial evidence indicates that supportive relationships are associated with higher levels of trait mindfulness [12, 26], whereas toxic social experiences such as peer victimization and discrimination are associated with lower levels of trait mindfulness [9, 12].

### Early adolescence as a transitional period

Early adolescence–the ages between 10 and 14 years—is characterized as a key transitional time in the life cycle; a time that provides a unique opportunity in which to study human development due to the quantity, nature, and speed in which changes occur. Indeed, rapid

changes occur across an array of domains and contexts in a relatively short time-span–physical changes, biological changes in relation to brain development and puberty, cognitive and emotional changes, school transitions, and changes in social relationships, including increasing importance and influence of peers and changes in the nature of parent-child and adult-child relationships [27]. There is evidence to suggest that although the role of peers in early adolescence increases significantly, family members–especially parents–continue to be an important source of support [28].

Transitional periods such as adolescence have been defined as phases in the life span in which developmental challenges and demands are intensified, and can be considered as phases of heightened "vulnerability and risk." Transitional periods have also been characterized as "transition-linked turning points" in development; events that have the potential to alter behaviour, affect, cognition, or context and can result in lifelong changes [29]. Taken together, understanding trait mindfulness and factors associated with it in early adolescence can provide insights into how it might be cultivated during this critical transitional period.

## Victimization and mindfulness

Peer victimization peaks between early and middle adolescence [30, 31] and poses critical challenges to the development of mindfulness. Peer victimization involves recurring experiences of social exclusion and/or direct physical, verbal, or cyber aggression, often at the hands of a more powerful peer [32, 33]. A robust literature documents victimization's pernicious long-term correlates in terms of adolescents' social, emotional, and academic functioning [34], and it has been theorized that victimization similarly undermines the development of mindfulness [9]. For example, a longitudinal study of 7th and 8th graders in the United States found that victimization predicted decreases in mindfulness across the 4-month study [9]. The authors of that study offered the explanation that peer victimization may repeatedly prompt worries about past or future bullying, detracting from adolescents' ability to attend to experiences emerging in the present moment. Such findings align with recent research using a person-centered approach to examine cross-sectional links between profiles of internet risk (which included metrics of victimization) and mindfulness, finding that a latent class of adolescents characterized by high scores on bullying victimization and perpetration (vs. a no-risk class) scored lower on the acting with awareness and nonjudging facets of mindfulness [35].

Several streams of related research are instructive in understanding the link between victimization and mindfulness. For example, research has found that the related construct of discrimination is negatively linked to mindfulness in high school [12] and college students [5]. Among potential psychological pathways, rumination may account for the victimization-mindfulness link, reflecting the assertion that victimization prompts worries about past or future bullying, thereby undermining present-centered awareness [9]. Research has found that adolescents who reported higher levels of victimization tended to report higher levels of rumination [36, 37], and rumination cross-sectionally and longitudinally predicted lower scores on several mindfulness facets during adolescence [2, 38–40]. Collectively, this web of empirical findings further suggests that victimization is a potential impediment to the development of mindfulness.

In short, evidence stemming from a single study on victimization and mindfulness as well as from studies examining correlates of victimization (i.e., discrimination and rumination) converge on the premise that victimization may undermine the development of mindfulness during early adolescence. The current study builds on prior research by testing longitudinal links from victimization to mindfulness across a 3-year interval.

## Supportive relationships and mindfulness

Beyond victimization, it is important to consider other relational components in adolescents' social ecologies, including supportive relationships with parents, teachers, and peers. Previous research indicates that supportive relationships seem to be particularly important during early adolescence [41, 42].

We argue that relationships characterized by high levels of trust, safety, and support may foster the development of mindfulness through at least two pathways. First, supportive relationships provide a secure, low-anxiety interpersonal context. In the presence of people who generally hold goodwill towards them, there is little reason to worry about how they will be treated or whether they are accepted, valued, or belong—at least with respect to those particular relational partners. In other words, supportive relationships are unlikely to elicit anxious responses that are at odds with mindfulness. Second, supportive relationship partners are more likely to show interest in one's emotions, thoughts, and needs, signaling that one's own emerging emotions, thoughts, and needs deserve attention and acceptance—the very hallmarks of mindfulness. Similarly, the reciprocity entailed in mutually supportive relationships calls for nonjudgmental awareness of *others'* emotions, thoughts, and needs as they are communicated in the present moment (i.e., interpersonal mindfulness) [43]. In other words, supportive relationships may invite enactments of mindfulness of self as well as others.

Few longitudinal studies have examined links between supportive relationships and mindfulness, but at least two are available. A recent longitudinal study across all 4 years of high school found that reports of mindful awareness were higher in years when adolescents reported higher needs-supportive relationships with their parents, teachers, and friends (i.e., supporting autonomy, competence, and relatedness) [12]. A second longitudinal study conducted across one school year found that high school students who perceived their teachers as more kind, clear, and calm (i.e., "mindful teaching") at the start of the year reported higher school need fulfillment at mid-year, which in turn predicted increased mindfulness at the end of the year [26]. Thus, initial evidence supports the view that supportive relationships are conducive to the development of mindfulness.

Evidence from cross-sectional studies further buttress this claim. For example, parents' reports of "mindful parenting" (which includes listening to one's children with full attention) were positively associated with self-reported mindfulness among adolescents [44–46]. In addition, evidence with college student samples found that social support was positively associated with mindfulness [47, 48], particularly when support came from friends [49] and family [49, 50]. Notably, one study found that the link between mindful parenting and adolescent mindfulness was mediated through attachment security [45], harkening back to the trust and safety pathway noted above. Indeed, a meta-analysis of cross-sectional studies with adult samples found that those with higher levels of either anxious or avoidant attachment were less mindful [51], and similar results have been found in the quality of adolescents' attachments to their parents [52, 53].

Supportive relationships may also *protect* adolescents who struggle with victimization. Although we are unaware of research examining buffering effects of supportive relationships when mindfulness is the outcome, supportive relationships buffer the negative associations between victimization and well-being outcomes. For example, early adolescent girls with high levels of victimization suffered less in terms of depressive symptoms—and had higher levels of life satisfaction and self-esteem—if they were strongly connected to both peers and adults [54]. In addition, in a year-long longitudinal study, both boys and girls who reported higher levels of support in their friendships displayed weaker prospective links between victimization and social emotional well-being [55]. The moderating role of supportive relationships has also been examined within-persons in a daily diary study [56]. On days when high school students spent time

with their friends, cyber victimization was not associated with anger, distress, or attendance problems, whereas cyber victimization *was* linked to these outcomes on days when they did not spend time with friends. Taken together, these findings suggest that supportive relationships with family and friends not only have longitudinal and cross-sectional links to mindfulness, but also moderate the negative association between victimization and mindfulness.

## The current study

We employ data from a large sample of Canadian young adolescents to cross-sectionally and longitudinally examine elements of adolescents' social ecologies that may play roles in the naturalistic variation in trait mindfulness and its development from 4th to 7th grade (the two timepoints at which data were available). Grounded in insights from dynamic theories of personality traits [14, 24], we argue that, in theory, trait mindfulness develops through recurring experiences likely to foster (or thwart) the repeated enactment of mindful states. Prior research suggests that victimization likely triggers rumination about past and future bullying, impeding the development of mindfulness [9], whereas supportive relationships with peers and adults likely conduce to mindfulness [12, 45] by providing safe, low-anxiety contexts where adolescents' emotions, thoughts, and needs are accepted and valued. We employ variable-centered and person-centered analyses that complement one another in providing a well-rounded portrait of the associations between social experiences and trait mindfulness.

Variable-centered analyses are instrumental in quantifying the effects of individual variables while controlling for potential confounds. We therefore employed structural equation model to test the hypothesis that (1) lower victimization and higher peer belonging and connectedness with adults at home will be associated with higher levels of mindfulness in 7th grade. In addition, based on research showing the moderating role of supportive relationships in buffering the negative association between victimization and adolescent well-being, we hypothesized that (2) the negative association between victimization and mindfulness in 7th grade will be weaker among those with higher (vs. lower) levels of peer belonging and connectedness with adults at home. Moreover, we hypothesized that (3) the effects expected above will be replicated longitudinally using the predictors from 4th grade and mindfulness from 7th grade.

Person-centered analyses are optimally suited to examine development from a holistic systems view [57], according to which complex patterns of interactive factors simultaneously operate within individuals' lives. Such a view aligns with ecological systems models of development [58] that imply the potential for heterogeneity in adolescents' social ecologies (e.g., high levels of victimization coexisting with high peer belonging for some adolescents). We therefore employ latent profile analysis (LPA), a person-centered approach, to accomplish the research goals to (4) explore common patterns in which victimization, peer belonging, and connectedness with adults at home manifest within children in 4th grade, and (5) examine whether class membership is associated with mindfulness in 7th grade. LPA and variable-centered moderation analyses share the goal of examining how combinations of variables function together, but rather than quantifying the effects of the variables LPA foregrounds subgroups of individuals with the most common patterns of scores observed in the data [59]. Thus, LPA is useful in describing the "kinds" of children and young adolescents that teachers, parents, and youth workers are most likely to encounter, potentially helping them identify which individuals may benefit most from additional resources and support.

## Materials and methods

### Participants

A total of 4,593 early adolescents were surveyed in 4th and 7th grade (50% girls; 50% boys; $M_{ageG4} = 9.02$, $SD = 0.24$; $M_{ageG7} = 12.19$, $SD = 0.54$). Students' first languages were English

(70.5%) followed by Mandarin (2.0%). Other first languages learned with more than 1% included Cantonese, Tagalog, Punjabi, and combinations of English and Tagalog, Punjabi, Mandarin, Spanish, and French. Participants were recruited from 291 elementary and middle schools in 32 public school districts in British Columbia, Canada.

All participants completed the Middle Years Development Instrument (MDI) [60], a population-level self-report survey of children's social and emotional development, physical health and well-being, and assets in the context of their home, school, and neighbourhood. Data from the MDI are aggregated for schools and communities and reported back in comprehensive reports and community maps to inform planning and decision making at local and regional levels. The MDI project began collecting data annually among 4th grade students in the 2009–10 school year for population monitoring purposes using brief measures for each construct. Three years later, the project was expanded for students in 7th grade. The 7th grade MDI survey contains all of the items in the 4th grade version as well as additional measures that were either considered more relevant to older children's experiences or were considered relevant for both grades but had to be cut from the 4th grade MDI survey because of length. As such, the present study draws from data collected as part of the larger MDI population level project that monitors early adolescents' experiences and well-being in different contexts—a project that was not specifically designed for the purposes of the current study. Data for the current study came from the cohorts of 4th grade students in 2013–14 and 2014–15 (7th grade in 2016–17 and 2017–18), as these are the only cohorts to-date for whom data were available on all study-related constructs.

A total of 2.3% of the data were missing. Of the original 4,593 cases, only 9 (0.20%) were omitted from the cross-sectional analysis, and 7 (0.15%) were omitted from the longitudinal analysis, as these cases had no data on any variables in their respective analyses. For the person-centered analyses, of the original 4,593 cases, 493 (10.7%) were omitted using current best practices for latent profile analysis with distal outcomes (described below). Due to the large proportion of missing data in the person-centered analysis, we conducted a sensitivity analysis in which only 75 (1.6%) of cases were omitted, and the same pattern of findings was obtained (described below).

## Procedure

The MDI is available to all public school districts in British Columbia. Among districts that elect to participate, administrators of individual schools and teachers decide whether to administer the MDI to their students. Passive consent is utilized, by which parents/guardians are notified of the research and must proactively decline to allow their children's participation. Students must positively assent to their own participation. On each page of the survey, students can discontinue. Average participation rates for participating school districts were high (2013–14 = 83%, 2014–15 = 84%, 2016–17 = 77%, 2017–18 = 82%). Note that the rate in 2016–17 was skewed by the 5% participation rate from a district that withdrew after 1 week of data collection. When this district was excluded from the computation, the participation rate was 83% for 2016–17. Given high participation rates from the 32 diverse participating districts, which included both rural and urban districts, the sample is fairly representative of the overall demographics of 4th and 7th grade students in British Columbia. Surveys were primarily administered in paper-pencil format when the 2013–14 cohort was in 4th grade, and primarily in electronic format when the 2014–15 cohort was in 4th grade. All 7th grade surveys were electronic. Differential item functioning analyses of MDI constructs have found no differences between the paper-pencil and electronic forms of the survey [61]. Surveys were administered within classrooms during school hours in November and December. To accommodate diverse

reading abilities among younger students, many teachers read the survey items aloud for their 4[th] grade students, whereas 7[th] grade students typically completed the survey silently by themselves. We do not know which students' teachers read the survey aloud, so we could not compare results along this dimension. This research was approved by the University's Behavioural Research Ethics Board as well as by the administration of each participating school district. Additional information on procedures involved with the administration of the MDI has been published elsewhere [60, 62].

## Measures

Please see the S1 Appendix for a complete list of items for all constructs. Because all of the measures were derived from the larger MDI survey, each construct entails three to five items selected on the basis of breadth of coverage by an interdisciplinary research team with expertise in each construct, focus group feedback from children, and in consultation with key stakeholders, including educators, parents/guardians, youth program providers, and community service organizations (e.g., United Way). As described elsewhere [60], the aim of the MDI was to create a comprehensive "whole child" survey that could be administered during one or two class periods to children in school, and that was developmentally appropriate and assessed multiple aspects of children's social and emotional development, physical health and well-being, and assets inside and outside of school. The majority of the scales on the MDI were derived from existing measures of relevant constructs, and the criteria for inclusion were (1) high reliability (i.e., Cronbach's alpha > .7) in previous research, (2) evidence of discriminant, convergent and content validity, and (3) age-appropriateness (age 9–12) of item content and wording. In order to create a survey that was both comprehensive and a manageable length to administer to students during class time, for scales with greater than three items, results of previously conducted factor analyses and reliability assessments (i.e., factor loadings, Cronbach's alpha after an item is deleted) were used to guide data reduction in order to shorten scales to a maximum of three to five items. For additional information on the development of the MDI and the validity of the scales, see [60, 62, 63].

**Trait mindfulness.** To align with recent conceptualizations of social and emotional learning (SEL), for the 7[th] grade MDI version measures designed to assess the dimension of "self-awareness" and related constructs (e.g., mindfulness) were reviewed following the process described previously. Three mindfulness items were identified for inclusion that were adapted for use with early adolescents from the Interpersonal Mindfulness in Teaching Scale [64], which has since been revised into the Mindfulness in Teaching Scale [65]. These items also have been used in a large-scale survey of elementary and high school students' social and emotional competencies by the American Institutes for Research (AIR) and the Collaborative for Academic, Social, and Emotional Learning (CASEL) [66]. Students used a 5-point scale (1 = *Disagree a lot*, 5 = *Agree a lot*) to respond to the items, "When I'm upset, I notice how I am feeling before I take action"; "When difficult situations happen, I can pause without immediately acting"; "I am aware of how my moods affect the way I treat other people". Arguably, these items capture the nonreactivity and acting with awareness aspects of mindfulness from the five-facet framework [16]. The mindfulness scale was administered in 7[th] grade only, and the items exhibited adequate internal consistency ($\alpha_{grade7}$ = .75).

**Peer victimization.** Four items adapted from the Safe School Student Survey [67, 68] measured physical, social, verbal, and cyber victimization. Students were provided with a definition of bullying, followed by the question, "This school year, how often have you been bullied by other students in the following ways?" Then, each of the four types of bullying was described—e.g., "Physical bullying (for example, someone hit, shoved, or kicked you, spat at

you, beat you up, or damaged or took your things without permission)"—followed by a
5-point response scale (1 = *Not at all this school year*, 5 = *Many times a week*). The four items
exhibited good internal consistency for both grades ($\alpha_{grade4}$ = .81; $\alpha_{grade7}$ = .79). In addition,
previous research has found that the peer victimization measure is negatively associated with
school belonging [60], peer belonging and caring school climate [63], and is positively associ-
ated with sadness and worries [63].

**Connectedness with adults at home.**    Three items from the California Healthy Kids Sur-
vey [69] measured connectedness with adults at home. Students used a 4-point scale (1 = *Not
at all true*, 4 = *Very much true*) to respond to the items (e.g., "In my home, there is a parent or
another adult who listens to me when I have something to say."). The three items exhibited
good internal consistency for both grades ($\alpha_{grade4}$ = .73; $\alpha_{grade7}$ = .82). In addition, previous
research has that the measure is positively associated with connectedness with adults at school,
with adults in the neighborhood, and with peers [60, 63], and is negatively associated with sad-
ness and worries [63].

**Peer belonging.**    Three items measuring peer belonging were adapted from the Relational
Provisional Loneliness Questionnaire [70]. Students used a 5-point scale (1 = *Disagree a lot*, 5
= *Agree a lot*) to respond to the items (e.g., "When I am with other kids my age, I feel I
belong."). The three items exhibited good internal consistency for both grades ($\alpha_{grade4}$ = .78;
$\alpha_{grade7}$ = .83). In addition, previous research has found that the peer belonging measure is posi-
tively associated with friendship intimacy [60, 63], caring school climate, optimism, self-con-
cept, and life satisfaction [63].

**Long-term self-regulation.**    Three items were used from the long-term subscale of the
Adolescent Self-Regulatory Inventory [71]. Since mindfulness was not measured in 4$^{th}$ grade,
this scale functioned as its proxy given its very strong association with mindfulness in 7$^{th}$
grade ($r$ = .95) at the latent variable level, and given the centrality of self-regulation as a constit-
uent element in certain conceptualizations of mindfulness [72]. This allowed us to pseudo-
control for previous levels of mindfulness in our longitudinal analyses. Students used a 5-point
scale (1 = *Disagree a lot*, 5 = *Agree a lot*) to respond to the items (e.g., "When I have a serious
disagreement with someone, I can talk calmly about it without losing control."). The three
items exhibited marginally adequate internal consistency ($\alpha_{grade4}$ = .67; $\alpha_{grade7}$ = .71).

**Gender and first language learned.**    Gender was collected through school records and
was coded 0 = *girl* and 1 = *boy*. Students' first language learned was measured with the item,
"What is the first language you learned at home? (*You can check more than one if you need
to*.)" Response options included Aboriginal Language, Cantonese, English, Filipino/Tagalog,
French, Hindi, Japanese, Korean, Mandarin, Punjabi, Spanish, Vietnamese, and Other. The
MDI has since evolved to measure gender identity and first language learned in more inclusive
ways (e.g., providing an option for describing one's gender "in another way" and using "First
Nations, Inuit or Métis" in lieu of "Aboriginal Language"). But here we report how these vari-
ables were measured among the cohorts for whom data were ready for analysis at the time of
this writing.

## Analytic plan

Data were first analyzed for missingness, as described in the Participants section. Descriptive
statistics and bivariate correlations among the study variables were next computed. Given that
students were nested within schools, we first examined the intraclass correlation (ICC) for
mindfulness in 7th grade. The ICC was .01, indicating that only 1% of the total variance in the
dependent variable (mindfulness) occurred between schools. In addition, the design effect
(DE) was 1.21, indicating that the nested design's violation of the independence assumption

had a negligible effect on standard error estimates. Since these statistics were well below the thresholds that require multilevel modeling (i.e., ICC < .05; DE < 2.0) [73], we did not specify commands that accounted for the nested data. Next, the substantive variable-centered and person-centered analyses were conducted. Please see the S2 Appendix for selected M*plus* scripts.

**Variable-centered analyses.** We tested our three hypotheses using structural equation modeling in M*plus* 8.3 [74], which uses full information maximum likelihood (FIML) to make use of information on all variables in the analysis to compute nonbiased parameter estimates and standard errors without unnecessarily omitting cases [75]. The robust maximum likelihood (MLR) estimator was used to adjust standard errors and chi-square values for non-normality in the data. The fixed factor method was used to identify the latent constructs [76], such that the variances of all latent variables were fixed to 1.0.

In the cross-sectional analysis of 7th grade data, a measurement model was specified first to examine model fit and factor loadings of the four latent constructs: mindfulness, victimization, connectedness with adults at home, and peer belonging. Individual items served as indicators of their respective latent constructs. Model fit was evaluated using the following thresholds for good fit: (1) a comparative fit index (CFI) > .95 for good fit and > .90 for adequate fit, (2) a Tucker-Lewis index (TLI) > .95 for good fit and > .90 for adequate fit, (3) a root-mean-square error of approximation (RMSEA) < .06 for good fit and < .08 for adequate fit, and (4) a standardized root-mean-square residual (SRMR) < .08 for good fit and < .10 for adequate fit [77]. The minimum threshold for "good" standardized factor loadings was .55 [78].

Next, a structural regression model was specified such that mindfulness was regressed on peer victimization, connectedness with adults at home, and peer belonging. Gender served as a covariate (we also ruled out gender as a moderator of the substantive links by constraining the paths to be equal across genders and found that doing so did not result in significant loss in model fit). Finally, the XWITH command in M*plus* was used to construct three two-way latent variable interactions (victim X adults, victim X peers, adults X peers), which were added as additional predictors of mindfulness in three separate models, thus limiting the potential for multicollinearity.

A similar procedure was followed for the longitudinal analysis, except 4th grade self-regulation served as an additional covariate (enabling us to pseudo-control for prior levels of mindfulness), and all predictors and interactions were from the 4th grade data. Mindfulness in 7th grade was the dependent variable.

Secondary versions of these analyses were conducted controlling for cohort, but the same general pattern of results was obtained, and the effect of cohort was non-significant in all cases. Therefore, we present results from analyses that excluded cohort from the models.

Statistically significant interactions were examined by testing simple slopes using the interaction tool available at jeremydawson.co.uk/slopes.htm. Simple slopes were evaluated at one standard deviation above and below the means of the moderating variables.

We report model fit statistics (CFI, TLI, RMSEA, SRMR) for the measurement and structural models for both the cross-sectional and longitudinal analyses. Then we report the standardized and unstandardized path coefficients, 95% confidence intervals, and p-values for each main effect and interaction effect. For simple slopes analyses we report the unstandardized coefficients, 95% confidence intervals, and p-values.

**Person-centered analysis.** Variable-centered analyses were complemented by latent profile analysis (LPA) in M*plus*. LPA identifies latent subpopulations (classes) in the data that have similar patterns of scores across a given set of observed variables [79]. Individuals within the same class share a relatively homogenous pattern of scores, whereas individuals from different classes display distinct patterns of scores. In our study, identification of classes was determined by patterns of scores on three relationship variables measured in 4th grade: peer

victimization, connectedness with adults at home, and peer belonging. Reflecting the ecological reality that these indicators share substantial variance, we specified covariance parameters among indicators (covariances were constrained to be equal across latent classes).

A variety of fit indices were used to select a model with the appropriate number of classes. In general, solutions with lower Akaike's Information Criteria (AIC), Bayesian Information Criteria (BIC), sample-size adjusted BIC (SSA-BIC), and higher entropy were preferred [79]. In addition, we employed the adjusted Lo-Mendell-Rubin likelihood ratio test (LMR) and the bootstrap likelihood ratio test (BLRT) to examine whether each solution's fit was a significant improvement over the solution with one less class [79]. Finally, models were preferred if they had a substantial number of cases in each class and if the solution was easily interpretable [79].

We report several indices in connection with model selection (AIC, BIC, SSA-BIC; LMR and BLRT tests; entropy, and size of smallest class) for each of five potential models ranging from one to five latent classes. Since the connectedness with adults at home measure used a different response scale than peer victimization and peer belonging, we used the proportion of scale maximum linear transformation to set all three indicators on the same 0–1 scale when reporting the final latent class solution. Each latent class was then labeled in a manner that characterized its combination of scores across the three social ecology indicators.

After selecting a model with an appropriate number of latent classes, class membership in 4th grade was used to examine mean differences across classes in mindfulness levels in 7th grade, controlling for 4th grade self-regulation and gender. This was accomplished using the manual BCH three-step method [80], which is among the current recommended approaches to examine distal outcomes of latent class membership while controlling for covariates [81]. Step 1 involves the computation of classification error for each individual's latent class membership, while specifying a model with the selected number of latent classes. Classification errors are then incorporated into Step 3 to appropriately weight the tests of mean differences in the distal outcome across latent classes (note that Step 2 is now subsumed by Step 3). Key advantages of this approach include incorporating classification error and minimizing shifts in latent class composition from Step 1 to Step 3. As mentioned above, M*plus* omitted a total of 493 cases (10.7%) from this analysis, either because they had no information on any of the latent class indictors in Step 1 ($n = 57$), or because they had at least some missing data on the covariates or outcome in Step 3 ($n = 436$). We display means, 95% confidence intervals for these means, standardized effect sizes (Cohen's *d*), and p-values in examining mean differences in mindfulness across latent classes.

Note that M*plus* treats both distal outcomes and covariates as exogenous variables in the BCH and similar procedures, and as such applies listwise deletion. This issue is an active area of inquiry. See, for example http://www.statmodel.com/discussion/messages/13/10373.html? 1582853519. As a sensitivity analysis that omits only a fraction of cases with missing data, we hard-classified cases based on the class in which each individual had the highest probability of membership, created dummy variables for each latent class, and used the dummy variables (along with self-regulation and gender) to predict 7th grade mindfulness in M*plus*. Missing data were handled in the model estimation process using FIML, and resulted in the omission of 75 cases (1.6%) that had no information on any of the analytic variables. This approach yielded the same pattern of findings as the recommended BCH procedure presented in the main text.

# Results

## Variable-centered analyses

We examined the roles of peer victimization, peer belonging, and connectedness with adults at home in early adolescents' mindfulness, both cross-sectionally in 7th grade and longitudinally from 4th to 7th grade. Correlations among the study variables are displayed in Table 1. Model

**Table 1. Correlations among study variables (*N* = 4,593).**

|  | 1 | 2 | 3 | 4 | 5 | 6 | 7 | 8 | 9 |
|---|---|---|---|---|---|---|---|---|---|
| 1. Mind G7 | --- |  |  |  |  |  |  |  |  |
| 2. Victim G4 | **-.11** | --- |  |  |  |  |  |  |  |
| 3. Victim G7 | **-.24** | **.32** | --- |  |  |  |  |  |  |
| 4. Adults G4 | **.17** | **-.14** | **-.09** | --- |  |  |  |  |  |
| 5. Adults G7 | **.37** | **-.12** | **-.25** | **.29** | --- |  |  |  |  |
| 6. Peers G4 | **.19** | **-.26** | **-.11** | **.42** | **.16** | --- |  |  |  |
| 7. Peers G7 | **.42** | **-.17** | **-.38** | **.19** | **.41** | **.31** | --- |  |  |
| 8. Self-reg G4 | **.27** | **-.12** | **-.07** | **.39** | **.18** | **.51** | **.20** | --- |  |
| 9. Self-reg G7 | **.95** | **-.14** | **-.27** | **.22** | **.43** | **.21** | **.46** | **.33** | --- |
| 10. Gender | **-.06** | .04* | -.01 | **-.11** | .00 | .01 | .04** | **-.08** | .00 |

Mind = mindfulness; Victim = victimization; Adults = connectedness with adults at home; Peers = peer belonging; Self-reg = self-regulation; Gender (0 = F, 1 = M).

* $p < .05$;

** $p < .01$; **bold values** $p < .001$.

fit for the measurement and structural models for both the cross-sectional and longitudinal analyses were good: CFI = .972 - .991, TLI = .964 - .988, RMSEA = .017 - .036, SRMR = .017 - .025. In addition, standardized factor loadings were reasonably high (.56 - .86).

In the cross-sectional regression analysis of the 7th grade data, as hypothesized, victimization had a small negative association with mindfulness and peer belonging and adult connectedness had moderate positive associations with mindfulness (Table 2). There was also a small 2-way interaction by which adult connectedness moderated the effect of victimization on mindfulness. Simple slopes analysis revealed a small but significant negative link between victimization and mindfulness at high values of adult connectedness (*B* = -0.14, *p* < .001, 95% CI [-.22, -.07]), but no link between victimization and mindfulness at low values of adult connectedness (*B* = -0.04, *p* = .209, 95% CI [-.11, .02]). Thus, contrary to our hypothesis, at high levels

**Table 2. Cross-sectional and longitudinal SEM regressing 7th grade mindfulness on social ecology factors.**

|  | Cross-Sectional: 7th Grade Predictors of 7th Grade Mindfulness (*N* = 4,584) | | | Longitudinal: 4th Grade Predictors of 7th Grade Mindfulness (*N* = 4,586) | | |
|---|---|---|---|---|---|---|
|  | *B* [95% CI] | *β* | *p* | *B* [95% CI] | *β* | *p* |
| *Main Effects* |  |  |  |  |  |  |
| **Self-reg** | -- | -- | -- | .24 [.17, .31] | .23 | < .001 |
| **Gender** | -.08 [-.11, -.04] | -.07 | < .001 | -.04 [-.07, .00] | -.04 | .034 |
| **Victim** | -.08 [-.13, -.03] | -.07 | .002 | -.07 [-.11, -.02] | -.07 | .002 |
| **Adults** | .27 [.21, .32] | .23 | < .001 | .05 [.00, .11] | .05 | .052 |
| **Peers** | .34 [.28, .40] | .30 | < .001 | .04 [-.03, .10] | .03 | .259 |
| *2-way Interactions* |  |  |  |  |  |  |
| **Victim * Adults** | -.05 [-.10, .00] | -.04 | .038 | -.01 [-.05, .04] | -.01 | .677 |
| **Victim * Peers** | -.01 [-.05, .04] | -.01 | .786 | -.04 [-.08, .00] | -.04 | .027 |
| **Adults * Peers** | .01 [-.04, .06] | .01 | .646 | .03 [-.02, .07] | .03 | .209 |

Dependent variable = 7th grade mindfulness; Self-reg = self-regulation; Gender (0 = F, 1 = M); Victim = victimization; Adults = connectedness with adults at home; Peers = peer belonging. Note that main effects are from models without 2-way interactions; 2-way interactions are reported from models testing each 2-way interaction separately.

of adult connectedness, the negative association between victimization and mindfulness was stronger than at low levels of adult connectedness (Fig 1).

Longitudinally, controlling for self-regulation in 4th grade, there was a small negative association between victimization in 4th grade and mindfulness in 7th grade, but 4th grade peer belonging and adult connectedness were not associated with mindfulness in 7th grade (Table 2). Of note is the fact that the longitudinal effect of connectedness with adults at home was non-significant in the planned main effects model ($p = .052$), but in all models that included 2-way interactions this effect achieved statistical significance, suggesting ambiguity as to the presence of this effect, as sometimes it was significant and sometimes it was not, depending on slight variations in the model. In addition, there was a small 2-way interaction by which peer belonging moderated the effect of victimization on mindfulness. Simple slopes analysis revealed a small but significant negative link between victimization and mindfulness at high values of peer belonging ($B = -.12$, $p < .001$, 95% CI [-.19, -.06]), but no link between victimization and mindfulness at low values of peer belonging ($B = -.04$, $p = .137$, 95% CI [-.09, .01]). Thus, contrary to our hypothesis, at high levels of peer belonging, the negative association between victimization and mindfulness was stronger than at low levels of peer belonging (Fig 2).

## Person-centered analysis

To further examine how the three relationship constructs function together in adolescents' lives, LPA of the 4th grade data augmented the variable-centered longitudinal findings. The goals of the LPA were to (a) explore common combinations of scores (across victimization, connectedness with adults at home, and peer belonging) that typify different fourth graders' social lives, and (b) examine whether these distinct relationship patterns are longitudinally associated with mindfulness in 7th grade. In examining solutions involving up to five latent classes, a 4-class solution generally fit the data better than solutions with fewer classes (Table 3), and a 5-class solution performed even better. However, the 5-class solution involved two very small classes (2.3% and 3.6% of cases), so the 4-class solution (Fig 3) was selected instead. Similar profiles for the 4-class solution were obtained using the final BCH 3-step procedure, and were labeled (1) *flourishing relationships* ($n = 2,990$; 72.9% of sample; low victimization, high connectedness with adults at home, high peer belonging), (2) *unvictimized but weak relationships with adults* ($n = 386$; 9.4% of sample; low victimization, low connectedness with adults at home, high peer belonging), (3) *moderately victimized but strong relationships* ($n = 561$; 13.7% of sample; moderate victimization, high connectedness with adults at home,

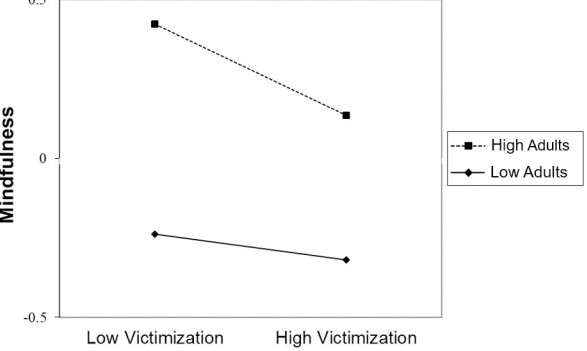

**Fig 1. Modeled mindfulness in 7th grade as a function of victimization and connectedness with adults at home in 7th grade ($N = 4,584$).** Adults = connectedness with adults at home.

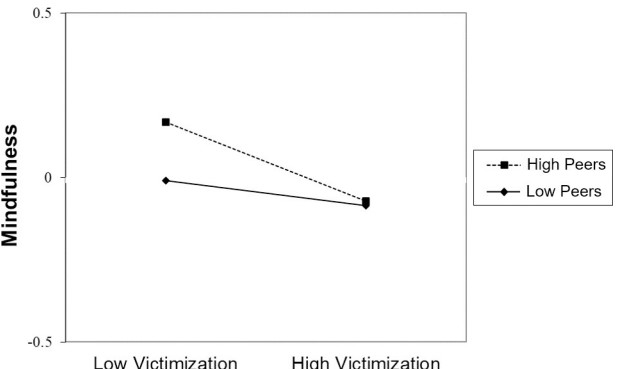

**Fig 2. Modeled mindfulness in 7th grade as a function of victimization and peer belonging in 4th grade (N = 4,586).** Peers = peer belonging.

high peer belonging), and (4) *victimized but strong relationships* (n = 163; 4.0% of sample; high victimization, high connectedness with adults at home, high peer belonging).

Next, we tested mean differences in 7th grade mindfulness across 4th grade latent classes, controlling for 4th grade self-regulation and gender. Results are displayed in Fig 4. Those who were in the *flourishing relationships class* in 4th grade had higher mindfulness in 7th grade than those in the *moderately victimized but strong relationships* class and the *victimized but strong relationships* class. In addition, those who were in the *unvictimized but weak relationships with adults* class in 4th grade had higher mindfulness in 7th grade than those in the *victimized but strong relationships* class. There were no other between-class differences.

## Discussion

Extending theory and research on naturalistic correlates of trait mindfulness during early adolescence, we cross-sectionally and longitudinally tested aspects of early adolescents' social ecologies in predicting trait mindfulness levels in 7th grade. In line with the premise that

**Table 3. LPA model selection criteria for victimization, connectedness with adults at home, and peer belonging among 4th graders (N = 4,536).**

|  | Model | | | | |
|---|---|---|---|---|---|
|  | **1-Class** | **2-Class** | **3-Class** | **4-Class** | **5-Class** |
| AIC | 30177.07 | 28253.89 | 27349.71 | 26749.81 | 26277.14 |
| BIC | 30234.85 | 28337.35 | 27458.85 | 26884.62 | 26437.64 |
| SSA-BIC | 30206.25 | 28296.04 | 27404.83 | 26817.89 | 26358.20 |
| Entropy | N/A | 0.94 | 0.89 | 0.88 | 0.89 |
| LMR | N/A | 1875.49*** | 885.88*** | 590.38*** | 466.80** |
| BLRT | N/A | 1931.17*** | 912.18*** | 607.91*** | 480.66*** |
| N for each class (%) | 1: 4536 (100) | 1: 4078 (89.9) | 1: 553 (12.2) | 1: 443 (9.8) | 1: 3149 (69.4) |
|  |  | 2: 458 (10.1) | 2: 3566 (78.6) | 2: 191 (4.2) | 2: 552 (12.2) |
|  |  |  | 3: 417 (9.2) | 3: 613 (13.5) | 3: 571 (12.6) |
|  |  |  |  | 4: 3289 (72.5) | 4: 102 (2.3) |
|  |  |  |  |  | 5: 162 (3.6) |

LPA = latent profile analysis; AIC = Akaike's Information Criteria; BIC = Bayesian Information Criteria; SSA-BIC = sample-size adjusted BIC; LMR = adjusted Lo-Mendell-Rubin likelihood ratio test; BLRT = bootstrap likelihood ratio test.

** $p < .01$.

*** $p < .001$.

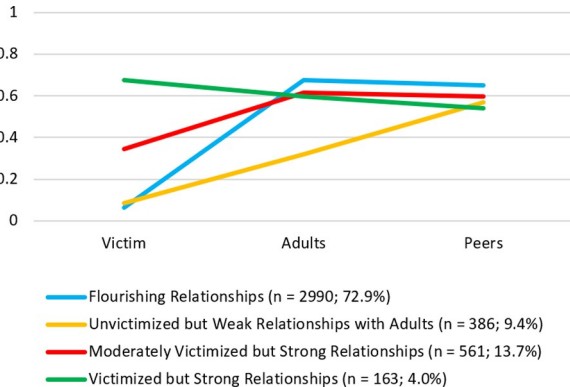

**Fig 3. Selected 4-class latent profile analysis solution and class proportions among 4th graders (*N* = 4,100).**
Victim = victimization; Adults = connectedness with adults at home; Peers = peer belonging. Given their different response scales, scores on each variable were linearly transformed onto the same 0–1 scale using the proportion of maximum scoring transformation.

mindfulness develops in contexts that are likely to foster (or thwart) repeated enactments of mindful states, we found that low victimization was linked to higher levels of trait mindfulness. This link was observed both cross-sectionally in 7th grade as well as longitudinally (controlling for 4th grade self-regulation) across a 3-year interval from 4th to 7th grade, indicating small but robust relational correlates of trait mindfulness during early adolescence. In addition, cross-sectionally (but not longitudinally), high peer belonging and connectedness to adults at home were each uniquely linked to higher levels of trait mindfulness in 7th grade, indicating compatibility among these constructs, but not necessarily pointing to a developmental process. Finally, contrary to our hypothesis, we found interactions indicating that supportive relationships may intensify the negative association between victimization and mindfulness.

## Victimization and mindfulness

We found that victimization in 4th grade was linked to slightly lower mindfulness in 7th grade, as well as cross-sectionally linked to mindfulness within 7th grade. These findings extend short-term longitudinal findings that victimization is linked to lower mindfulness [9], by demonstrating that this link persists across 3 years of development. Establishing this longitudinal link opens the door to understanding the processes by which it operates, and the means by which it can be disrupted. Below, we attend to each of these issues in turn.

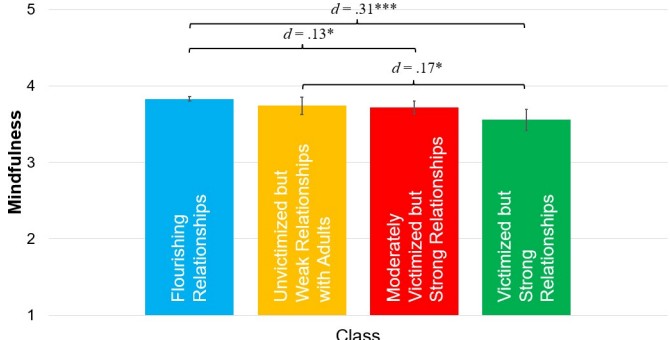

**Fig 4. Mean differences in 7th grade mindfulness across 4th grade latent classes, adjusting for 4th grade self-regulation and gender (*N* = 4,100).** Error bars are 95% confidence intervals; *d* = Cohen's *d*. *p < .01. ***p < .001.

Bullying and victimization are particularly salient in the early adolescent years [30, 31], and the most effective preventions programs entail systemic, "whole school" change [82]. Our findings suggest that such efforts may not only decrease victimization and thereby promote well-being and academic outcomes, but may also support the development of mindfulness, a key competency that contributes to positive functioning broadly speaking [1, 83]. Additionally, whole-school prevention programs that emphasize the importance of teacher-student relationships and cooperative learning structures [84] are likely to foster supportive relationships with teachers and peers, which our research (in part) and others' [26] indicate may pay further dividends in the cultivation of adolescents' mindfulness.

Establishing the longitudinal link from victimization to mindfulness also opens discussion into the pathways that explain this process. Given other research findings on the role of rumination in both victimization and mindfulness [36, 40], rumination is a strong explanatory candidate. Future research should examine rumination as a potential mediating pathway by which victimization undermines the development of mindfulness. In other words, victimized adolescents may either stagnate or decrease in trait mindfulness because they are more likely to ruminate upon past and future incidents of victimization. The heightened self-consciousness [85, 86] and concern about social evaluation [87] that are characteristic of early adolescence may particularly implicate rumination during this developmental period. In addition, given that mindfulness interventions reduce rumination [88–90], future research should examine the efficacy of mindfulness training in disrupting this potentially pernicious process. Future research should also consider the possibility that the link from victimization to mindfulness flows in the opposite direction. It could be the case that mindful youth are less likely to be victimized by their peers, perhaps because they are less reactive and tend to have stronger interpersonal skills.

## Supportive relationships and mindfulness

Beyond victimization, we tested the roles of two assets in adolescents' social ecologies. As hypothesized, cross-sectionally, both peer belonging and connectedness to adults at home were moderately linked to higher mindfulness in 7th grade. These findings corroborate past research showing that supportive relationships are positively linked to mindfulness during adolescence [12, 26, 45], and they are consistent with broader research showing that social support and parent-child attachment are positively linked to mindfulness [45, 47–53]. However, longitudinally, there were no links from 4th grade levels of these assets to mindfulness in 7th grade.

Future research should employ longitudinal study designs with more frequent measurements of supportive relationships and mindfulness to examine the developmental process across shorter time intervals. Social ecologies with peers and adults at home change considerably in the transition from middle childhood to early adolescence, and habituated levels of mindfulness are likely more strongly tied to their current contexts than to conditions that occurred years earlier. Such future research should also examine potential pathways by which supportive relationships may foster the development of mindfulness. One possibility—and consistent with our cross-sectional findings—is that supportive relationships provide adolescents with enduring access to low-anxiety contexts characterized by high levels of trust and safety, which may broadly conduce to present-centered awareness. A complementary possibility is that in supportive relational contexts—particularly those with high levels of emotional intimacy—adolescents and their social partners are more likely to nonjudgmentally attend to each other's thoughts, emotions, and needs, signaling that one's own emerging experiences as well as the communicated experiences of others are worthy of acceptance and awareness.

These pathways should be tested against viable competing (and perhaps complementary) explanations that examine the causal flow in the opposite direction, such as the possibility that mindfulness conduces to positive interpersonal functioning, for example, by supporting adolescents' ability to attend to peers' and adults' thoughts, emotions, and needs.

## Combined functions of victimization and supportive relationships

**Variable-centered findings.** Beyond testing the main effects, we examined the combined functions of victimization and supportive relationships using both variable- and person-centered approaches. With respect to variable-centered analyses, longitudinally we found that peer belonging moderated the link between victimization and mindfulness, although not in the manner we hypothesized. Contrary to our hypothesis, we found that the negative association between victimization and mindfulness was slightly *stronger* among those with high levels of peer belonging, whereas victimization was not linked to mindfulness at low levels of peer belonging. A similar pattern was observed cross-sectionally, but connectedness with adults at home instead moderated the link between victimization and mindfulness. Specifically, the negative association between victimization and mindfulness was slightly *stronger* among those with high levels of connectedness to adults at home, whereas victimization was not linked to mindfulness at low levels of connectedness to adults at home.

Contrary to our hypothesis, these findings suggest that victimization may have a stronger harmful effect for adolescents who have good relationships with peers and adults at home. In other words, diverging from research showing a buffering effect of supportive relationships on the impact of victimization on well-being outcomes [54–56], our findings suggest that supportive relationships may intensify the negative association between victimization and mindfulness. However, it is also worth noting that these interaction effects were very small, and that intermittent effects for different relational groups (peers vs. adults) were implicated longitudinally versus cross-sectionally, decreasing confidence in the replicability of the moderating roles of supportive relationships.

Thus, tentatively, perhaps since adolescents who have strong relationships tend to be the most mindful, they simply have the greatest room to fall when they are victimized. Alternatively, adolescents with supportive relationships may generally expect others to treat them with kindness and respect, and peer victimization violates these expectations, prompting a more intense reaction. By implication, teachers and youth workers should not discount the harmful effects of victimization on adolescents who otherwise have a strong relational support system.

It is also important to clarify that, despite the nuance offered by the interactions, our data do not indicate that peer belonging or connectedness to adults at home are harmful in any way. On the contrary, adolescents who thrived in terms of both kinds of supportive relationships reported higher levels of mindfulness cross-sectionally.

**Person-centered findings.** A person-centered approach linking 4th graders' relationship profiles to their mindfulness in 7th grade augmented our variable-centered findings. Person-centered analyses align with a "whole child" orientation to development by acknowledging complex interactive factors that co-exist within a child's life [57]. LPA identifies subgroups of individuals with similar patterns of scores across the variables of interest. We examined common patterns of victimization, peer belonging, and connectedness with adults at home, and found four latent classes of 4th graders.

A total of 72.9% of 4th graders in our sample were in the *flourishing relationships* class. This class had higher mindfulness scores in 7th grade than the two classes that had moderate or high levels of victimization and slightly lower (but still rather strong) relationships with adults and peers. These findings indicated that the majority of 4th graders were flourishing in terms of

their relationships with peers and adults at home, they experienced low levels of victimization, and 3 years later reported the highest levels of mindfulness. In other words, functioning at the highest level across all three elements of social ecology in 4th grade was associated with higher mindfulness as they transition into early adolescence.

However, close inspection of the person-centered findings suggested that victimization in particular drove between-class differences. Considering that there were only slight differences with respect to connectedness to adults at home and peer belonging between the *flourishing relationships* class (blue line), *moderately victimized but strong relationships* class (red line; 13.7% of sample), and *victimized but strong relationships* class (green line; 4.0% of sample), it is evident that victimization was the primary differentiating factor of these classes, and their mindfulness levels in 7th grade directly followed victimization levels in 4th grade. Indeed, the one class that diverged by having relatively weak connectedness to adults at home (*unvictimized but weak relationships with adults* class; yellow line; 9.4% of sample) nevertheless had similar levels of mindfulness in 7th grade as the *flourishing relationships* class, and despite having weak connectedness with adults at home still outperformed the *victimized but strong relationships* class in terms of mindfulness in 7th grade. Collectively, the person-centered findings tell a story that corroborates the variable-centered findings: Victimization–but not connectedness to adults at home or peer belonging–in 4th grade seems to be the key factor (among those examined) associated with mindfulness levels in 7th grade. Together, these findings underscore the need for teachers, youth workers, and parents to emphasize the importance of treating peers in inclusive and respectful ways that minimize and obstruct peer victimization.

## Limitations

Several caveats contextualize our findings. First, mindfulness was not measured in 4th grade and we instead used a measure of self-regulation as a proxy to control for prior levels of mindfulness in our longitudinal analyses. As a result, we cannot fully rule out the possibility that the observed longitudinal association between victimization in 4th grade and mindfulness in 7th grade may be due to shared variance between victimization and mindfulness in 4th grade. This concern is somewhat allayed by the high degree of shared variance between the self-regulation and mindfulness measures in 7th grade (i.e., when both measures were available), suggesting that self-regulation was a viable proxy in attempting to control for previous levels of mindfulness. Nevertheless, future longitudinal research should employ the same measure of mindfulness across measurement waves.

Second, our mindfulness measure was derived from a scale of teachers' mindfulness. However, it is noteworthy that the same items have since been used in children's surveys by the American Institutes for Research (AIR) and the Collaborative for Academic, Social, and Emotional Learning (CASEL) [66], suggesting that the mindfulness items are regarded as developmentally appropriate adaptations to the teacher measure. Moreover, the mindfulness scale had adequate internal consistency and was associated with constructs within our study in expected ways.

Finally, we note that our underlying rationale—that social ecologies that repeatedly prompt (or thwart) the enactment of mindful states result in the consolidation of mindful (or mindless) traits over time—was not explicitly tested. Indeed, the 3-year time interval between measurement occasions was not ideally suited to detect the longitudinal roles of social ecological factors, particularly during a time in the lifespan when young people's social ecologies undergo significant changes. Future research should employ experience sampling, daily diaries, and other intensive intraindividual measurement techniques to more sensitively examine the roles of victimization and supportive relationships in triggering mindful states, and test whether growth in mindful states over time accounts for growth in trait mindfulness [13].

## Conclusion

Our findings contribute to a growing body of evidence indicating that trait mindfulness is associated with ecologically normative experiences in adolescents' everyday lives. Relational contexts in particular have emerged as fertile soil in accounting for natural variation in trait mindfulness. Our research documents that adolescents who experience low levels of victimization in 4th grade are more mindful in 7th grade, and adolescents who experience low levels of victimization and high levels of supportive relationships with peers and adults within 7th grade are concurrently more mindful. These findings bear an important insight on a parallel stream of research pointing to trait mindfulness as a protective factor in the context of victimization [3, 4, 6]. Namely, the adolescents who might benefit most from trait mindfulness—victimized youth with weak relationship supports—are the least likely to have it. Both mindfulness-based interventions and systemic efforts to change adolescents' social ecologies have critical roles to play in nurturing a more mindful next generation.

## Supporting information

**S1 Appendix. Complete list of items for all constructs.**
(DOCX)

**S2 Appendix. M*plus* scripts.**
(DOCX)

## Acknowledgments

We would like to thank the MDI team at the Human Early Learning Partnership, and the school administrators, teachers, and especially students throughout British Columbia for making this research possible.

## Author Contributions

**Conceptualization:** Michael T. Warren, Kimberly A. Schonert-Reichl.

**Formal analysis:** Michael T. Warren, Randip Gill.

**Funding acquisition:** Kimberly A. Schonert-Reichl.

**Methodology:** Michael T. Warren, Anne M. Gadermann, Eva Oberle.

**Project administration:** Michael T. Warren.

**Supervision:** Michael T. Warren.

**Validation:** Michael T. Warren.

**Visualization:** Michael T. Warren.

**Writing – original draft:** Michael T. Warren.

**Writing – review & editing:** Kimberly A. Schonert-Reichl, Randip Gill, Anne M. Gadermann, Eva Oberle.

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
