## [Decision Letter · Decision Letter 0]

14 Oct 2020

PONE-D-20-21484

Naturalistic development of trait mindfulness: A longitudinal examination of victimization and supportive relationships in early adolescence

PLOS ONE

Dear Dr. Warren,

Thank you for submitting your manuscript to PLOS ONE. After careful consideration, we feel that it has merit but does not fully meet PLOS ONE’s publication criteria as it currently stands. Therefore, we invite you to submit a revised version of the manuscript that addresses the points raised during the review process.

The paper has several strengths including a large sample and longitudinal data. We were fortunate to get two experienced reviewers and I have read the paper as well. The reviewers did an excellent job describing the concerns and limitations of the paper. I will not reiterate each of their points, but rather, I will highlight several aspects of the paper that require your attention.

The paper requires major revisions, particularly regarding the data analyses. If you decide to revise and resubmit the paper, there is not guarantee that it will be accepted even with the changes.

Please provide a clearer theoretical rationale for conducting the LPA (Reviewer 1). Reviewer 2 suggested extracting the classes using grade 4 data to predict changes in mindfulness from grade 7, controlling for grade 4.

Reviewer 2 had several important questions about the psychometrics of the mindfulness measure used. For example, what is the validity of the Interpersonal Mindfulness in Teaching Scale for use in a sample of children? What were the psychometric characteristics of the specific mindfulness facets used?

Both reviewers noted concerns about the shortened / nonstandard measures used. What is the evidence of the validity of these shortened versions of the measures? Reviewer 1 suggested conducting a factor analysis of the item level data to see if the measures are behaving as they should.

Both reviewers expressed concerns about the data analyses and suggested several alternative approaches including using an SEM framework and testing moderation through multigroup analysis (Reviewer 2). Please also address the concerns of Reviewer 1 regarding how the tests of the interactions were conducted.

Finally, please address the questions raised by Reviewer 2 about the 3-year time frame.  

We look forward to receiving your revised manuscript.

Kind regards,

Judy Garber, Ph.D.

Academic Editor

PLOS ONE

Journal Requirements:

Reviewers' comments:

Reviewer's Responses to Questions

**Comments to the Author**

1. Is the manuscript technically sound, and do the data support the conclusions?

Reviewer #1: Partly

Reviewer #2: Yes

2. Has the statistical analysis been performed appropriately and rigorously? 

Reviewer #1: No

Reviewer #2: Yes

3. Have the authors made all data underlying the findings in their manuscript fully available?

Reviewer #1: Yes

Reviewer #2: Yes

4. Is the manuscript presented in an intelligible fashion and written in standard English?

Reviewer #1: Yes

Reviewer #2: Yes

5. Review Comments to the Author

Reviewer #1: The present manuscript investigates the development of mindfulness in early adolescence. Using a very large dataset (N=4593) involving 32 different schools, individuals were assessed in 4th grade and again in 7th grade. The authors found that lower victimization and greater belonging and connectedness with adults had higher levels of mindfulness years late. Person-centered analyses also identified four classes of mindfulness. The topic is very interesting, and the sample size is impressive. I offer the following critiques of the manuscript in its present form:

1) The tests of the interactions are not interpretable in their current form. When testing for a three-way interaction, it is necessary to include all underlying two-way interactions (just as seen in Table 2). However, those tests of the two-way interactions in their second step are not interpretable in this analysis. Each controls for the other, and the collinearity of XY, ZY, ZX interaction terms will be very high, and will greatly 'overcontrol' these tests. Instead, these two-way interactions should be tested in three separate models, each examining one of these two-way interactions. My guess is that this is what led to the unexpected observed effects.

2) The assessment of these constructs is of course a limitation of the study because of shortened / nonstandard measures. Some of these items, though, look like they might not really measure the constructs in the study. I'm particularly thinking about the relation between the mindfulness items and the long-term self-regulation items. This would likely be a concern for most readers. I would suggest subjecting the item-level data to a factor analysis, so that it is possible to really see if these measures are behaving as they should.

3) I wasn't really sure if the LPA analyses added much to the manuscript. It looks like classes were extracted from grade 7 data, and then associated with the grade 7 mindfulness measure. So it's really just a correlational finding. It would be more interesting if the classes were extracted using grade 4 data, and then entered into an analysis of predicting changes in mindfulness from grade 7, controlling for grade 4 mindfulness.

Reviewer #2: This study examined features of adolescents’ social ecologies that support or hinder the naturalistic development of trait mindfulness. Results indicated that lower victimization, higher peer belonging, and higher connectedness with adults at home predicted higher trait mindfulness between 4th and 7th grade. Moderation tests revealed mixed evidence that, contrary to hypotheses, connectedness with adults exacerbated the negative effect of peer victimization on mindfulness. Person-centered analyses yielded different categories of social ecologies, and that adolescents in “flourishing” ecologies had the highest levels of trait mindfulness. Strengths of the study include large sample size and longitudinal data collection. Below are areas of concern about the manuscript and analyses.

Introduction

The introduction did not adequately set up the latent profile analysis. Though ample space was devoted to setting up the variable-centered hypotheses, the LPA was tacked onto the end of the introduction and almost felt like an afterthought. There seem to be unique strengths associated with using a person-centered approach, in that they provide a high-level portrait of the overall social ecologies that might differentiate trait mindfulness. Are there prior studies (even with adults) that take this approach? If so, what do results reveal? The authors also suggest a practical implication for conducting LPA (for identifying adolescents who might benefit from additional support), but what are the theoretical reasons for conducting LPA?

Method

The sample includes 4593 students from over 300 elementary and middle schools. To what degree is the sample representative of the overall demographics of British Columbia?

The authors should justify more why they assessed adolescents’ mindfulness using the Interpersonal Mindfulness in Teaching Scale (which presumably was created to measure classroom teachers) rather than validated scales (e.g., MAAS, CAMS). Is there data to suggest this scale is a valid measure of mindfulness in this sample? Have other studies used this scale to measure mindfulness in youth? Moreover, categorizing specific items into mindfulness facets (observing, non-reacting, acting with awareness) seems a bit of a stretch. Is this how the original IMTS treats these items? Also, prior research has shown that the observing facet does not function well psychometrically in non-meditating samples of adults and adolescents. Combining an item that purportedly captures “observing” with items that purportedly capture other facets (e.g., nonreactivity) could impact the scale’s validity.

Along these same lines, can the authors provide further rationale for how they pick items from each of the other scales (e.g., peer belonging)? How many items did the original scales contain? Has prior work demonstrated that the shortened versions of each scale maintain construct validity? Can the authors also provide a full list of the items used in an appendix? This will help to evaluate the full coverage of each construct.

Results

The authors mention that 2.3% of the data were missing overall. However, the information in Table 2 suggests a higher percentage of missing data (e.g., Longitudinal analyses show N=3,945, but with a full sample of 4,593, the missing rate would be closer to 14%). Can the authors explain these discrepancies?

For longitudinal analyses, a stronger analytic approach would be to conduct an autoregressive, cross-lagged panel analysis using all the variables measured at each time point. Such a model would also allow for modeling covariances within time-points. As such, I would encourage the authors to also include 7th grade self-regulation as a covariate of 7th grade mindfulness (it seems right now that they only include 4th grade self-regulation). Using a SEM framework would also permit the use of full information maximum likelihood to retain missing data (see above point).

Table 2 should include more information, including unstandardized regression coefficients, 95% CI, and exact p-values.

The authors do a nice job setting up the moderation analyses and are transparent in reporting the results (since they do not align with hypotheses). However, I might suggest exploring whether gender moderates the associations between social ecology factors and mindfulness. For example, does peer belonging predict mindfulness more strongly for one gender vs. the other? If the authors adopt a SEM framework, I might encourage them to test moderation through multi-group analysis for all moderation tests.

Discussion

Can the authors explain more in the discussion, and throughout the paper, why they used a 3-year time frame? How might this timeline impact the interpretation of results? For example, are social ecologies expected to have some stability to impact mindfulness 3 years later? Or would there be intervening steps (e.g., high adult connectedness leads to greater trust and safety) that could produce higher mindfulness even in shifting social ecologies? Other limitations that might need to be acknowledged would be the use of short-form versions of scales designed to measure constructs of interest.

6. PLOS authors have the option to publish the peer review history of their article (what does this mean?). If published, this will include your full peer review and any attached files.

Reviewer #1: No

Reviewer #2: No

---

## [Author Response · Author response to Decision Letter 0]

15 Dec 2020

Please find attached a Word document titled "Response to Reviewers."

---

## [Decision Letter · Decision Letter 1]

24 Feb 2021

PONE-D-20-21484R1

Naturalistic development of trait mindfulness: A longitudinal examination of victimization and supportive relationships in early adolescence

PLOS ONE

Dear Dr. Warren,

Thank you for submitting your manuscript to PLOS ONE. After careful consideration, we feel that it has merit but does not fully meet PLOS ONE’s publication criteria as it currently stands. Therefore, we invite you to submit a revised version of the manuscript that addresses the points raised during the review process.

The paper has several strengths including a large sample and longitudinal data. We were fortunate to get experienced reviewers and I have read the paper as well. The reviewers did an excellent job describing the concerns and limitations of the paper.

The first reviewer was satisfied that you had addressed the concerns outlined in the first review. The second reviewer had several comments that need your attention. With some rewriting, the issues likely can be addressed. I will not reiterate Reviewer 2's comments, but I request that you address each of them in your response. The paper requires modest revisions, particularly regarding the interpretation of the findings and eliminating all language that implies causality. If you decide to revise and resubmit the paper, there is no guarantee that it will be accepted even with the changes.

.Please submit your revised manuscript by April 15, 2021. If you will need more time than this to complete your revisions, please reply to this message or contact the journal office at plosone@plos.org. Please include the following items when submitting your revised manuscript:

We look forward to receiving your revised manuscript.

Kind regards,

Judy Garber, Ph.D.

Academic Editor

PLOS ONE

Journal Requirements:

Reviewers' comments are available in the attachment and below.

This study examined correlates of trait mindfulness in students in grades 4 and 7. Strengths of the study include a large sample, high participation rates, longitudinal design, and the use of two different data analytic approaches. The following concerns should be addressed:

1. The primary concern about this paper is how the authors frame their study and interpret their findings. All references to causal language should be removed and rather, the findings should be reframed in terms of correlations and associations rather than using terms such as “longitudinal development,” the “role” of, “foster,” “impede.” No causal conclusions can be made from these data. Only studies in which randomized controlled trials that intervene to change mindfulness can draw conclusions about causes of increases in trait mindfulness. Please remove all causal language. Several examples follow.

p. 3 “longitudinal development of trait mindfulness during early adolescence”

p. 4 “Ecologies that are likely to repeatedly enable mindful states may contribute to the development of trait mindfulness”

p. 4 “Initial evidence indicates that supportive relationships may foster trait mindfulness

[12,26], whereas toxic social experiences such as peer victimization and discrimination may

impede the development of trait mindfulness [9,12].”

P. 4 “relatively few developmental periods in which so many changes occur in a relatively short

time-span.” Birth through age five? Although the authors state that there are relatively few developmental periods, I suggest that the authors change this to indicate that this is an important time period to see changes but not that it is any more or less critical than another such period as birth to age 5.

P. 5 “understanding trait mindfulness and the factors that promote or impede its development in early adolescence can provide insights into how it can be cultivated during this critical transitional period.”

P. 5. victimization “undermines the development of mindfulness”

P. 6 “potential role of victimization in mindfulness.”

P. 6. “victimization as an impediment to the development of mindfulness”

P. 6 “constructs closely related to victimization (i.e., discrimination and rumination) converge on the premise that victimization may undermine the development of mindfulness during early adolescence.” What do you mean by “closely related?” Are these correlates or similar in terms of their meaning? They might correlate with rumination, but neither victimization nor discrimination are the same as rumination.

P. 7 “Supportive relationships are particularly impactful during early adolescence”

P. 7. Supportive relationships “foster the development of mindfulness through at least two pathways”

P. 9 “Taken together, these findings suggest that supportive relationships with family and friends not only show promise in fostering the development of mindfulness, but also in diminishing the detrimental effect of victimization on mindfulness.”

P. 9 “the roles of social experiences in the development of trait mindfulness.” This study does not tell us about the “roles” of social experiences in the “development” of trait mindfulness. Rather, it looked at correlates of change.

P. 24. “indicating a small but robust relational process by which trait mindfulness develops during early adolescence.”

P. 25. “Our findings suggest that such efforts may not only decrease victimization and thereby promote wellbeing and academic outcomes, but may also support the development of mindfulness.”

P. 26 The authors state that “there were no links from 4th grade levels of these assets to mindfulness in 7th

grade, suggesting that supportive relationships do not foster the development of mindfulness.” This is a big leap. It is not possible to conclude anything about whether supportive relationships foster the development of mindfulness. The authors should simply report that finding of a lack of a significant association without suggesting anything from this null finding.

P. 27. Regarding the unexpected significant interaction, the authors state that “the pernicious effect of victimization on mindfulness was slightly stronger.” Again, the results do not show that victimization had an effect on mindfulness. Rather, the significant interaction should be interpreted as showing that the association between victimization and mindfulness varied as a function of level of connectedness with adults. The authors do state this correctly on P. 22 (Figure 2) “contrary to our hypothesis, at high levels of peer belonging, the negative association between victimization and mindfulness was stronger than at low levels of peer belonging.”

P. 29. The authors’ description of the results of the person-oriented analyses are stated correctly: “functioning at the highest level across all three elements of social ecology in 4th grade was associated with higher mindfulness as they transition into early adolescence.

P. 29. Victimization – but not connectedness to adults at home or peer belonging – in 4th grade seems to be the key factor (among those examined) explaining mindfulness levels in 7th grade.” Victimization is related to mindfulness, but it does not explain or cause levels of mindfulness in 7th grade.

P. 30. Although the discussion of anti-bullying programs is interesting, it does not follow from the findings of the current study.

P. 31. “Our findings contribute to a growing body of evidence indicating that trait mindfulness

develops as a function of ecologically normative experiences in adolescents’ everyday lives.”

A second major concern involves the measures of the constructs.

a. Three items measured most of the constructs. Although the internal consistencies were reported, more information about the psychometrics of these “measures” should be provided in this paper, even if it has been reported elsewhere.

b. Given that there are only three items per construct, all three items should be included in the description of the measures or in a table so that they can be evaluated.

c. Two of the three items on the mindfulness scale seem to be measuring impulsivity rather than mindfulness characteristics such as awareness or focusing on the present.

2. Another measurement concern is that mindfulness was not measured in 4th grade. The authors state that the measure of long-term self-regulation is a pseudo proxy for the mindfulness items measured in grade 7, but it is not clear that they are justified in this assertion. Mindfulness and self-regulation are highly correlated in 7th grade, but the correlations of 4th grade self-regulation and 7th grade mindfulness are not strong, which might indicate that they are not really the same construct. This is a major concern in this paper.

The authors assert that “self-regulation was a viable proxy” for mindfulness. This statement is not sufficiently justified.

3. Given the few findings regarding the relations among victimization, parental support, and mindfulness, the authors should provide some explanation for their finding that:

“supportive relationships intensified the adverse effect of victimization on mindfulness.”

   a. The authors suggest that: “peer victimization may repeatedly prompt worries about past or future bullying, detracting from adolescents’ ability to attend to experiences emerging in the present moment.” Although this is one possible explanation, wouldn’t this also be the case for any kind of worries, not just victimization by peers? Is there something about victimization in particular that would be linked to mindfulness?

   b. The logic of these potential connections makes sense, as long as it is not presented as either causal or primary. Many different types of stressors could be associated with mindfulness besides victimization. Is it victimization per se or anything that leads to rumination?

   c. In the Introduction (p. 6), the authors speculate about several ways the constructs might be related, but they do not test these suggestions. For example, “rumination may account for the victimization-mindfulness link, reflecting the assertion that victimization prompts worries about past or future bullying, thereby undermining present-centered awareness [9]. Research has found that adolescents who reported higher levels of victimization tended to report higher levels of rumination [36,37], and rumination cross-sectionally and longitudinally predicted lower scores on several mindfulness facets during adolescence.”

P. 7. “in the presence of people who generally hold goodwill towards them, there is little reason to worry about how they will be treated or whether they are accepted, valued, or belong”

P. 7 “supportive relationships are unlikely to elicit anxious responses that are at odds with mindfulness.”

Although these are reasonable speculations, they might be more relevant in the discussion rather than the introduction. Did they have a measure of rumination that they could include in the analyses?

   d. The authors also should consider the possibility of the direction of the association between victimization and mindfulness goes the other way. Is it possible that youth who are more mindful are less likely to be victimized by peers? The authors do note possible reverse directionality (p. 27) regarding the relation between mindfulness and positive peer relationships.

Other points to address:

1. p. 9. Was there a developmental reason for the two ages at which mindfulness was studied other than that these were “the two timepoints at which data were available?” “these are the only cohorts to-date for whom data were available on all study-related constructs.”

2. Explain more about the larger study and sample from which these data are a part. Make it clear that this study involved the analysis of data collected as part of another study and that the larger study was not specifically designed for this purpose.

3. P. 10. Hypothesis 2: “in the presence of higher (vs. lower) levels of peer belonging and connectedness with adults at home, the negative association between victimization and mindfulness in 7th grade will be weaker.” Will be weaker than what or as compared to what?

4. Were there any differences in the findings for the sample that used paper vs. online administration in 4th grade? Also, were there differences between the results for children whose teachers read the surveys to them vs. did not?

5. Please provide all 3 items of connectedness and peer belongingness, self-regulation

6. Of the three items that measured mindfulness, the first two seem to be about impulsivity rather than mindfulness:

“When I’m upset, I notice how I am feeling before I take action.”

“When difficult situations happen, I can pause without immediately acting.”

“I am aware of how my moods affect the way I treat other people.”

7. Please included a more detailed description of the data analysis plan

8. Gender was a covariate, but did they test interactions with gender?

9. Please try to link the findings from the two different data analytic approaches.

10. Include the N for each class in Table 3

11. P. 24. “Extending theory and research on the naturalistic development of trait mindfulness duringadolescence, we cross-sectionally and longitudinally tested aspects of early adolescents’ social

ecologies in explaining naturalistic variation in trait mindfulness from 4th to 7th grade.”

Please change this first sentence. This study does not extend research on the naturalistic “development” of trait mindfulness during adolescence [4th grade to 7th grade is not really adolescence; maybe early adolescence]. Given that they did not measure trait mindfulness in 4th grade, despite their using the measure of self-regulation as a “pseudo proxy” for mindfulness, it was not, and therefore they should reframe their language about this.

Reviewers' comments:

Reviewer's Responses to Questions

**Comments to the Author**

1. If the authors have adequately addressed your comments raised in a previous round of review and you feel that this manuscript is now acceptable for publication, you may indicate that here to bypass the “Comments to the Author” section, enter your conflict of interest statement in the “Confidential to Editor” section, and submit your "Accept" recommendation.

Reviewer #1: All comments have been addressed

Reviewer #3: (No Response)

2. Is the manuscript technically sound, and do the data support the conclusions?

Reviewer #1: Yes

Reviewer #3: No

3. Has the statistical analysis been performed appropriately and rigorously? 

Reviewer #1: Yes

Reviewer #3: Yes

4. Have the authors made all data underlying the findings in their manuscript fully available?

Reviewer #1: Yes

Reviewer #3: Yes

5. Is the manuscript presented in an intelligible fashion and written in standard English?

Reviewer #1: Yes

Reviewer #3: Yes

6. Review Comments to the Author

Reviewer #1: (No Response)

Reviewer #3: The authors need to revise their interpretation of their findings using language that does not imply causality.

7. PLOS authors have the option to publish the peer review history of their article (what does this mean?). If published, this will include your full peer review and any attached files.

Reviewer #1: No

Reviewer #3: No

---

## [Author Response · Author response to Decision Letter 1]

24 Mar 2021

Please see the attached file labeled, "Response to Reviewer."

---

## [Editor Report · Decision Letter 2]

19 Apr 2021

Naturalistic development of trait mindfulness: A longitudinal examination of victimization and supportive relationships in early adolescence

PONE-D-20-21484R2

Dear Dr. Warren,

We’re pleased to inform you that your manuscript has been judged scientifically suitable for publication and will be formally accepted for publication once it meets all outstanding technical requirements.

Kind regards,

Judy Garber, Ph.D.

Academic Editor

PLOS ONE

Additional Editor Comments (optional):

Thank you for attending to the Reviewers' comments and for making the appropriate changes in the manuscript. We appreciate your responsiveness and believe that the manuscript is much improved as a result.

---

## [Editor Report · Acceptance letter]

26 Apr 2021

PONE-D-20-21484R2 

Naturalistic development of trait mindfulness: A longitudinal examination of victimization and supportive relationships in early adolescence 

Dear Dr. Warren:

I'm pleased to inform you that your manuscript has been deemed suitable for publication in PLOS ONE. Congratulations! Your manuscript is now with our production department. 

Kind regards, 

on behalf of

Dr. Judy Garber 

%CORR_ED_EDITOR_ROLE%

PLOS ONE